# Efficient Urban Inundation Model for Live Flood Forecasting with Cellular Automata and Motion Cost Fields

**Maikel Issermann [1]** , **Fi-John Chang [1,\*]** **and Haifeng Jia [2,\*]**

[1]   Department of Bioenvironmental Systems Engineering, National Taiwan University,
     Taipei City 10617, Taiwan; d04622003@ntu.edu.tw

[2]   Division of Environment System Analysis, School of Environment, Tsinghua University,
     Beijing 100084, China

\*   Correspondence: changfj@ntu.edu.tw (F.-J.C.); jhf@tsinghua.edu.cn (H.J.)

**Abstract:** The mitigation of societal damage from urban floods requires fast hydraulic models for emergency and planning purposes. The simplified mathematical model Cellular Automata is combined with Motion Cost fields, which score the difficulty to traverse an area, to the urban inundation model CAMC. It is implemented with simple matrix and logic operations to achieve high computational efficiency. The development concentrated on an application in dense urban built-up areas with numerous buildings. CAMC is efficient and flexible enough to be used in a "live" urban flood warning system with current weather conditions. A case study is conducted in the German city of Wuppertal with about 12,000 buildings. The water depth estimation of every time step are visualized in a web-interface on the basis of the virtual globe NASA WorldWind. CAMC is compared with the shallow water equations-based model ANUGA. CAMC is approximatively 5 times faster than ANUGA at high spatial resolution and able to maintain numerical stability. The Nash-Sutcliffe coefficient (0.61), Root Mean Square Error (0.39 m) and Index of Agreement (0.65) indicate acceptable agreement for water depth estimation but identify different areas where important deviations occur. The estimation of velocity performs considerably less well (0.34 for Nash-Sutcliffe coefficient, 0.13 ms$^{-1}$ for Root Mean Square Error, and 0.39 for Index of Agreement) because CA ignores momentum conservation.

**Keywords:** urban inundation model; cellular automata; motion costs; efficient simulation; virtual globe; ANUGA

## 1. Introduction

Climate change and urbanisation are driving forces in the increasing occurrence and severity of urban floods [1–4]. A famous example of the devastation which an urban flood can cause is the flash flood of 21 July 2012 in Beijing [5,6]. An extreme rainfall event with a maximum of around 460 mm caused the death of 79 people and destroyed over 8000 buildings. The total value of property damage was approximately 10 billion RMB (1.6 billion USD).

Such tragedies underline the importance of continuing efforts in the development of urban inundation models [7]. Rescue and emergency responders could better plan relief operations if inundation models could provide high-resolution forecasts under current weather conditions in a very short time [8,9]. In addition, urban floods can be mitigated by smoothing peaks and slowing flows with appropriate stormwater management. Fast and accurate inundation models would benefit the corresponding planning processes by facilitating the application of spatial optimization [10,11]. This is especially pertinent for the cost-effective planning of low impact development (LID) to develop

a sustainable urban stormwater system [12,13]. In the context of efficient and sustainable resource utilization, stormwater should also be perceived as a valuable resource [14,15]. Its harvesting and integration with other urban environmental systems could lead to a reduced ecological footprint of urbanization.

However, the routing methods of most two-dimensional physical models (e.g., ANUGA [16,17]) rely on the solution of the shallow water equations (SWE), in which mass and momentum are conserved [18]. Numerical techniques (e.g., finite element and finite volume) are implemented to solve this complex partial differential equation system by iterative computation. Such implementation is very taxing for computer systems. Model runtime is roughly proportional to the reciprocal of grid resolution to the power of three [19]. Optimization with such models can be challenging and requires many computational resources depending on the resolution. Their low computational efficiencies restrict their usages for emergency flood forecasts under current weather conditions and demand ongoing research efforts.

The simplification of hydraulic processes is a common strategy in the development of more efficient models. Assumptions, such as the ignorance of inertia or the uniformity of flows, are employed to simplify the shallow water equations. The products are the diffusive (e.g., LISFLOOD-FP [20]) and kinematic wave equations (e.g., HEC-HMS [21]), respectively. Another strategy is to improve the implementation of inundation models. Parallelization and the utilization of graphic processing units (GPU) in hydraulic modelling [22–24] experienced an increase in recent years. The improvement of computational processes is particularly advanced in commercial development. Furthermore, machine learning triggered a revived interest in the application of statistical models for flood modelling [25–27]. Although such models effectively provide real-time estimations [28] and are particularly relevant for emergency responders [29], the regularly necessary training of such models to adjust them to the changing urban fabric constrains their usages. Therefore, physical models would still be needed for the continual adjustment to changing conditions. Machine learning depends on the quality and quantity of training data [30]. The adjustment to a changing environment also limits the applicability of machine learning algorithms as an inundation model in the planning of stormwater management.

In this study, hydraulic processes were simplified to develop an efficient urban inundation model called CAMC. CAMC is a realization of the principles of cellular automata (CA) [31] in combination with Motion Cost fields (MC), which is a computing technique in the field of crowd pathfinding and steering [32]. CA is a low-level abstraction of physical processes focusing on the conservation of mass. It can be classified as "finite volume" implementation [18] because it discretizes the spatial domain into control volumes (e.g., regular 2D cells). Previous studies and models on the basis of CA proved its accuracy in modeling hydraulic processes and its computational efficiency [33–36]. CA models behave like diffusive wave-like models [35] by neglecting the conservation of momentum. This neglect and the way of attributing discharge lead to the inherent weakness of accurately estimating velocity vectors that provide the Cartesian coordinates, and thus the incidence angles, of water flows.

This study describes the development of a CA model that relies on simple matrix and logic operations, while avoiding the relatively computationally expansive Manning's equation [33,35] and iterative sorting algorithms [34] as an integral component of its CA formulation. This implementation is simpler and still able to control oscillation which is a common ailment of CA models. This advantage allows for the simulation of urban inundation with high spatial and temporal resolution. According to Bernardini et al. [37], compact urban fabrics increase the risks for individuals in the case of urban flooding. CA models have the prospect to supply time-sensitive urban flood warnings under diverse situations (e.g., pipe leakage, river overflow, spatially variable rainfall). Flood-induced evacuation strategies would thus be better informed. In spite of findings that the modeling of buildings affects flow characteristics [38], many CA studies focus on watershed-scale applications and use in their validations "bare-earth" urban areas devoid of buildings. It poses a missed opportunity not to dedicate CA for urban inundation modeling because CA is capable to deal with spatial high-resolution data in an

acceptable amount of time. The impact of spatial resolution on the estimation of flow characteristics [39] emphasizes the significance. In this study, much effort and care was spent on the preparation of a case study in the German city of Wuppertal with a dense urban fabric of about 12,000 buildings with the intention to compare the model performance of a CA with an SWE model. Besides modeling buildings as flow obstacles, CAMC is even equipped to incorporate the drained water from roofs of buildings. The Motion Cost fields are not only the basis for the attribution of discharge between the cells, but also represent an interface to integrate surface features and soil properties with the aim to incorporate infiltration and evapotranspiration. However, this work focuses on the model description and the performance comparison with the SWE model ANUGA. In order to ensure comparability between ANUGA and CAMC, infiltration and evapotranspiration are ignored in the simulation because ANUGA was not developed for such application. The short-term objective is the provision of a fast flood model for urban flood warning. CAMC's simpler model formulation and pure implementation in Python allows for its integration into a simulation framework for urban environmental impact, called WupperWWEM [40], and demonstrate its applicability and efficiency. It provides "live" flood warnings on the basis of current weather conditions and visualizes the flood extent of every time step in the web-interface of WupperWWEM. The long-term objective is to apply CAMC for the planning and spatial optimization of urban runoff source control to achieve a sustainable urban stormwater system.

## 2. Methodology

The following section describes the modelling approach and system architecture of the urban inundation model CAMC, which is incorporated in the "live" urban simulation WupperWWEM [40].

### 2.1. Cellular Automata

The underlying flow routing method in CAMC rests on the principles of cellular automata (CA). CA are a simple mathematical system grounded on self-organization and discretization of space and time [31]. This system consists of three fundamental elements: cell neighbourhoods, cell states and state transition rules.

The spatial dimension in this implementation of CA is discretized into square grid cells. Other forms of tessellation, such as triangular [41] and hexagonal [42,43], are possible. Each cell is embedded in a local vicinity of cells, called cell neighborhoods. Common definitions of cell neighborhoods are the Moore and the von Neumann neighborhoods [44]. In the Moore neighborhood, all adjacent cells are considered (with a total of 8 cells), while the von Neumann neighborhood is diamond-shaped with four cells at each edge (with a total of 4 cells). In CAMC, the von Neumann definition is utilized.

Another fundamental element is the cell states. Each cell has a state of water depth. A positive influence on the water depth is exerted by existing water depth, inflows from neighboring cells, rainfall and water drained from roofs. Diminishing factors on the cell state are outflows to neighboring cells, infiltration and evaporation. The determination of water depth in the central cell is given by Equation (1).

$$H_{i,t} = H_{i,t-1} + \left( \sum D_{ji,t-1} - \sum D_{ij,t-1} \right) \frac{\Delta t}{b^2} + \left( R_t - E_t - I_t + NumBldg \times RD_t \right) \Delta t \qquad (1)$$

where $H_{i,t}$ is the current water depth in the central cell $i$ (L); $D_{ji,t-1}$ is the previous discharge from neighbouring cells to central cell $i$ (inflow; ($L^3 T^{-1}$)); $D_{ij,t-1}$ is the previous discharge from the central to the neighbouring cells (outflow; ($L^3 T^{-1}$)); $b$ is the cell width (L); $R_t$ is the current rainfall intensity (L/T); $E_t$ is the current evaporation rate (L/T); $I_{i,t}$ is current the infiltration rate (L/T); $NumBldg$ is the number of cells indicating buildings; $RD_t$ is the current rate of roof drainage (L/T).

## 2.2. Motion Costs

The transition rule stipulates how the cell state evolves over time, and thus causing the self-organising behavior exhibit by CA. The global application of the transition rule in combination with a simplified spatial discretization leads to an inherent advantage in computational efficiency.

In this implementation of CA, the global transition rule is derived from Motion Cost fields (MC). MC are the basis in the model formulation of Flow Field tiles, which are an efficient and flexible computing technique in the field of crowd pathfinding and steering. Thus, it is increasingly deployed in robotics and computer games. MC discretize the simulation domain into a grid, where each cell has a numeric property. These values represent the path cost for a traversing agent [32]. The costs are varying depending on terrain features and properties (e.g., slopes and roughness). Values can be assigned to signal impassable obstacles. Hence, MC reflect the difficulty, and thus the energy effort, for an agent to traverse an area. The MC are globally derived and shared with all agents. The dynamic of agents can be defined to follow an energy-minimization or maximization paradigm.

In this model, discrete volumes of water manifest the agents. MC are variable depending on terrain features (e.g., roughness length and ground elevation), existing water depth and sinks, i.e., infiltration. The schema of MC is shown in Figure 1. The schema also illustrates the possibility of extending the model to subsurface flows.

Ground elevation has the greatest impact in order to delineate the influence of gravity. Roughness length (e.g., vegetation) also contributes to the increase of MC. In contrast, the infiltration capacity of the soil negatively influences MC with the aim of simulating the impact of sorptivity of soils. But its influence diminishes over time because of increasing saturation. The MC are time-varying because water depth and infiltration capacity change over time. Buildings are categorically prohibited from receiving inflows. MC are calculated by Equation (2).

$$MC_{i,t} = h_i + H_{i,t} + k_i - \left[ \alpha D_{usS} M_i - \sum_{t=0}^{t} I_{i,t} \Delta t \right] \tag{2}$$

where $MC_{i,t}$ are the current Motion Costs of the central cell $i$; $h_i$ is the elevation (L); $H_{i,t}$ is the current water depth (L); $k_i$ is the roughness length (L); $\alpha$ is the fraction of infiltration potential; $D_{usS}$ is the initial depth of unsaturated soil (L); $M_i$ is the moisture deficit; $I_{i,t}$ is the infiltration rate (L/T). The discharge of the central cell is distributed according to the MC, which serve as weights. The model follows an energy-minimizing paradigm, and thus water preferably flows to cells with the lowest MC. In a physical sense, water flows follow the law of gravity and are mainly driven by the difference of elevation between the cells.

Figure 2 elucidates with a numerical example of how discharge from the central cell to the neighboring cells is calculated by the means of MC. Cells with equal or higher MC than the central cell $i$ are excluded from the discharge attribution because of the energy effort required to overcome gravity. The MC of all other cells, including the central cell, are summed up (Step 1) and the fractions from total MC are formed (Step 2). The MC fraction of each considered cell is converted to the reciprocal (Step 3) and the sum of the reciprocals form the denominator for the gain fraction. The reciprocals lead to the share of discharge (Step 4), which is called the gain fraction. The gain fraction determines how much of the water depth in the central cell is attributed to the considered cells (Step 5). The gain fraction describes the relative attractiveness of a cell's properties for inflow, which is mainly determined by the elevation of the cell. Unlike previous CA implementations, the central cell is included in the calculation and can retain water. For the rare cases that the discharge will lead to excess water depth in a neighboring cell compared to the central cell, the excess water depth will be shared with the central cell. This mechanism serves to minimize oscillation, which is a common problem in CA models. The entire computation is implemented with efficient matrix and logic operations. It avoids the utilization of the computationally intensive Manning's equation [35] and iterative sorting algorithms for cell ranks [34].

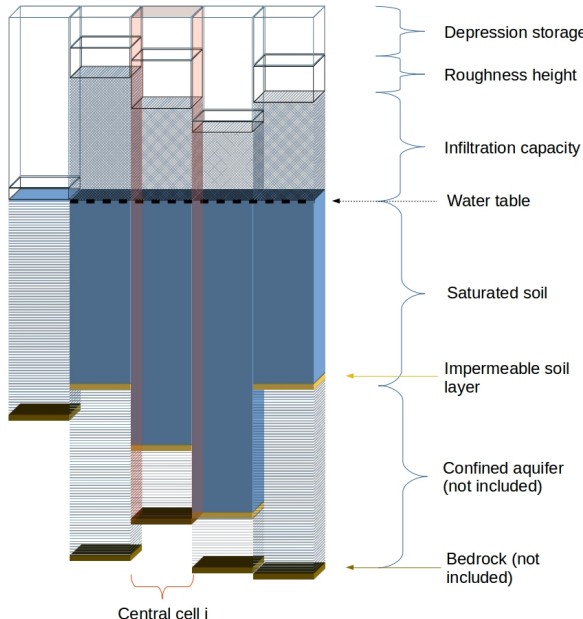

**Figure 1.** Schema of motion costs in a von Neumann neighbourhood.

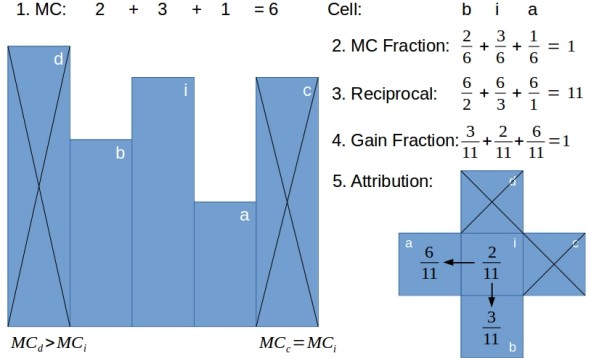

**Figure 2.** Step-by-step example for the calculation of discharge.

### 2.3. System Architecture

The implementation of CAMC with the numerical Python library NumPy allows us to further ameliorate the computational efficiency of the model. Figure 3 exhibits the algorithm of CAMC in its "live" simulation mode. Before the start of the time loop, the simulation domain is defined and populated by importing the raster data of the Digital Elevation Map (DEM) and land cover data. In the case that infiltration should be considered, additional data on the soil types is required. Besides serving as an obstacle to runoff, buildings are additionally incorporated by considering the water drained from roofs in the computation of water depth. In the preparation of the simulation domain, cells are defined and receive water from buildings. One way is to declare all adjacent cells as receiving. Another way is to declare cells with a specific feature as receiving. For instance, the lowest cell adjacent to a building can be declared as receiving the drained water. Consequently, this cell will receive additional water and potentially exacerbate hotspots. Furthermore, the roof drainage has a time-delaying effect because of the sizing of the drainage system.

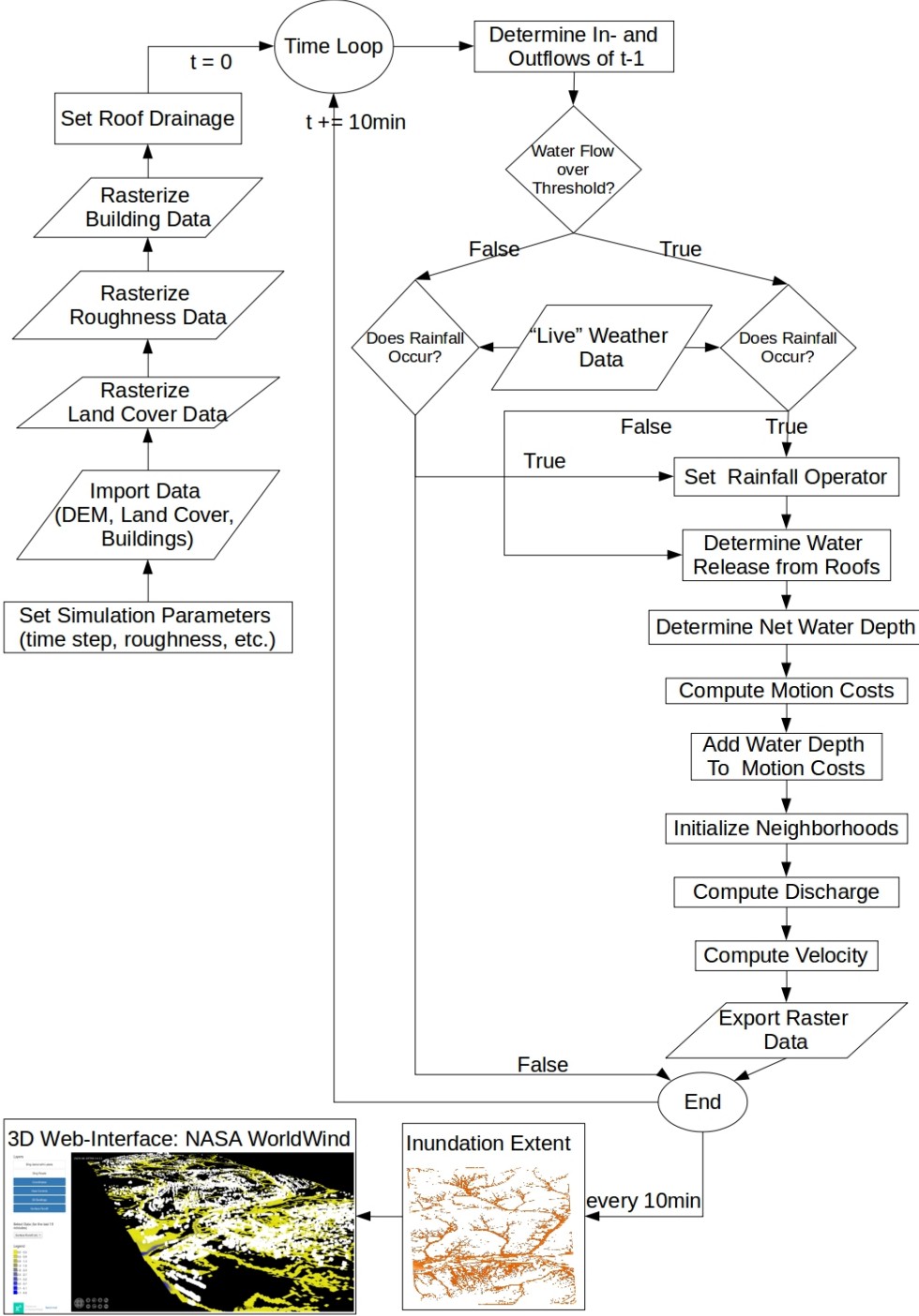

**Figure 3.** The algorithm of CAMC in "live" simulation mode.

The time loop starts with the determination of the in- and outflows of the central cell from the previous time step. Subsequently, the current weather data is retrieved and inquired whether rainfall occurs or not. If rainfall occurs, a rainfall operator will be set. The operator is the external force that evolves the cell states. The rainfall operator can be a single value, i.e., rainfall intensity, that is uniformly applied over the whole simulation domain. Alternatively, rasterized rainfall data describing the spatial distribution of rainfall intensity can be applied. Another option is the simulation of a leakage from a single or a group of cells. If no rainfall occurs, the setting of a rainfall operator is skipped. To complete the identification of the incoming water amount, the water drained from roofs is quantified according to the roof area and its drainage sizing. In addition to outflows,

evapotranspiration and infiltration can be calculated by the Hargreaves-Samani [45] and the Philip formula [46], respectively. In the next step, the net water depth is determined by accounting for all sources and sinks. The conservation of mass in the system is guaranteed. Therefore, all information for the computation of MC according to Section 2.2 are available. The cell neighborhoods containing the numerical values of MC are gathered and the discharge from the central cell is calculated with respect to the ratios of MC in the vicinity (Figure 2). The final step is the estimation of the flow velocity with the Gauckler–Manning equation [47] in Equation (3).

$$V = \frac{1}{n}Hr^{\frac{2}{3}}S^{\frac{1}{2}}$$

(3)

where $V$ is the velocity (L/T); $n$ is the Manning's n roughness value (T/L$^{\frac{1}{3}}$); $Hr$ is the hydraulic radius (L); $S$ is the friction slope (L/L).

The water depth is exported as raster data. The data is then uploaded to the web-interface of CAMC. The web-interface is based on the JavaScript library NASA WorldWind [48]. NASA WorldWind is a virtual globe that allows us to visualize geographic data.

## 3. Case Study

A case study was conducted with an urban area in the German city of Wuppertal. LiDAR data with a resolution of 4–10 points/m$^2$, which is provided from the geodata portal of the German federal state North Rhine–Westphalia [49], was utilized to generate a DEM with a resolution of 6 m$^2$. Data on buildings and green space was retrieved from OpenStreetMap [50]. The area of the simulation domain encompasses almost 10 km$^2$. The area is comprised of 25% green space and 17% building area, totaling around 12,000 buildings. The default land cover is paved. The two-dimensional maps of DEM and land cover are displayed in Figure 4. From the DEM, it is apparent that the area is a valley with hills to the north and south. Those hills contain most of the green space. The elevation difference is around 150 m with the maximal elevation at 304 m. A river, the Wupper, flows from East to West on the valley floor. The build-up area is concentrated along the river banks.

Two simulations were performed: a "live" simulation for urban flood warning and simulation for model comparison. The flowchart of the "live" simulation, which refers to simulations under current conditions, is shown in Figure 3. Current weather data is retrieved every 10 min from OpenWeather [51]. If rainfall is detected, the runoff flows are simulated. Unfortunately, OpenWeather only provides a uniform rainfall intensity for the whole area.

In the model comparison, runoff flows as estimated by CAMC are compared the results from the hydrodynamic model ANUGA, which was developed by the Australian National University (ANU) and Geoscience Australia (GA) [16]. ANUGA uses triangular tessellation and a finite volume approach to solve the shallow water equations. The model was validated against the analytical solution of the shallow water wave problem [52], as well as in a large benchmarking exercise of hydraulic modeling tools organized by the UK Environment Agency [18]. The simulation duration is 1 h with a 10 s time step. A rainfall event is simulated with an intensity of 400 mm/h and duration of 3 min. Buildings in ANUGA and CAMC were modeled as elevated areas in order to function as obstacles. Infiltration and evapotranspiration were ignored to ensure better comparability. The boundary conditions in both models were set to reflective, which does not allow the water to leave the simulation domain. Reflective boundary conditions ensure the conservation of mass in the system, whereas transmissive boundary conditions can create numerical instability in ANUGA [53]. The default Manning's n value, which refers to paved surfaces, was 0.02 s/m$^{\frac{1}{3}}$, whereas the friction of green space was 0.15 s/m$^{\frac{1}{3}}$. The gauge locations marked in Figure 4 serve to compare the temporal dynamics of both simulations.

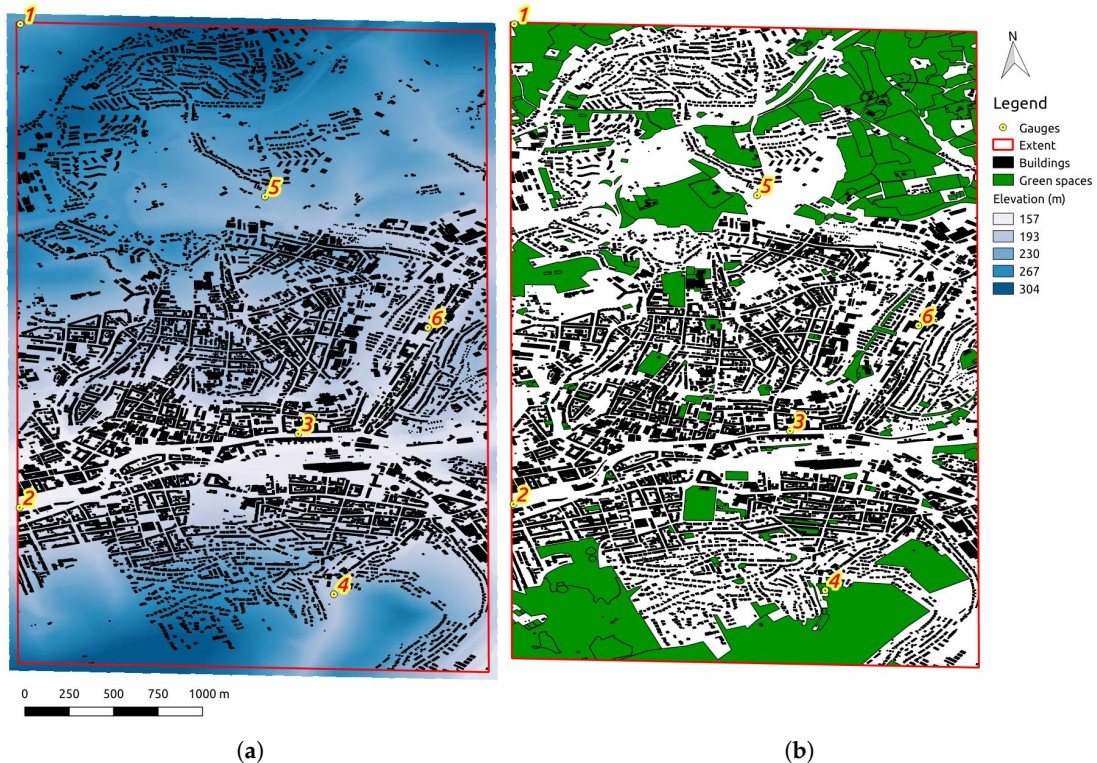

(**a**) (**b**)

**Figure 4.** Case study data for an urban area in Wuppertal: (**a**) DEM; (**b**) Land cover.

## 4. Results and Discussion

The objective of this work is to develop an urban inundation model that is efficient enough to supply "live" urban flood warning for a densely built-up area under current weather conditions. The output of one-time step can be seen as a 3D web visualization in Figure 5. It is part of the urban environmental monitoring system WupperWWEM [40].

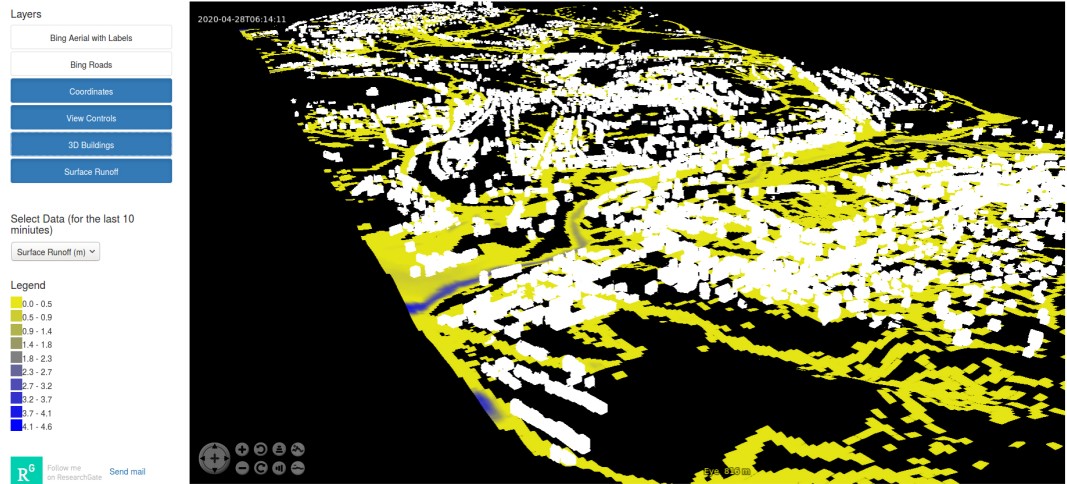

**Figure 5.** The 3D web-interface WupperWWEM for the visualization of "live" flood warning in an urban area of Wuppertal [40].

Regarding the validation study, Figure 6 illustrates the final water depth estimated by CAMC. It shows that the majority of the area, especially the elevated cells, were very quickly dried because the limited amount of runoff was drained into the depression. The road system served as a flow route towards the lower-lying cells where a river is located. The major hotspot was the western outflow of the river with a maximal inundation of almost 4.6 m. Gauge location 2 is located there. Other

principal hotspots were at gauge 1 in the north-western corner of the simulation domain and at gauge 5 on the northern hill slopes.

**Table 1.** Model efficiency performance measures.

| Name | Formula | Range | Ideal Value |
|------|---------|-------|-------------|
| Nash-Sutcliffe Model Efficiency (NSE) | $1 - \dfrac{\sum_{t=1}^{n}(y_{s,t}-y_{r,t})^2}{\sum_{t=1}^{n}(y_{r,t}-\overline{y}_r)^2}$ | $(-\infty, 1)$ | 1 |
| Root Mean Square Error (RMSE) | $\sqrt{\dfrac{1}{n}\sum_{t=1}^{n}(y_{s,t}-y_{r,t})^2}$ | $(0, \infty)$ | 0 |
| Index of Agreement (IoA) | $1 - \dfrac{\sum_{t=1}^{n}(y_{s,t}-y_{r,t})^2}{\sum_{t=1}^{n}(|y_{s,t}-\overline{y}_r|+|y_{r,t}-\overline{y}_r|)^2}$ | $(0, 1)$ | 1 |

$n$ is the number of time steps; $y_{s,t}$ is the simulated output at time step $t$;

$y_{r,t}$ is the reference output at time step $t$; $\overline{y}_r$ is the mean of the reference output

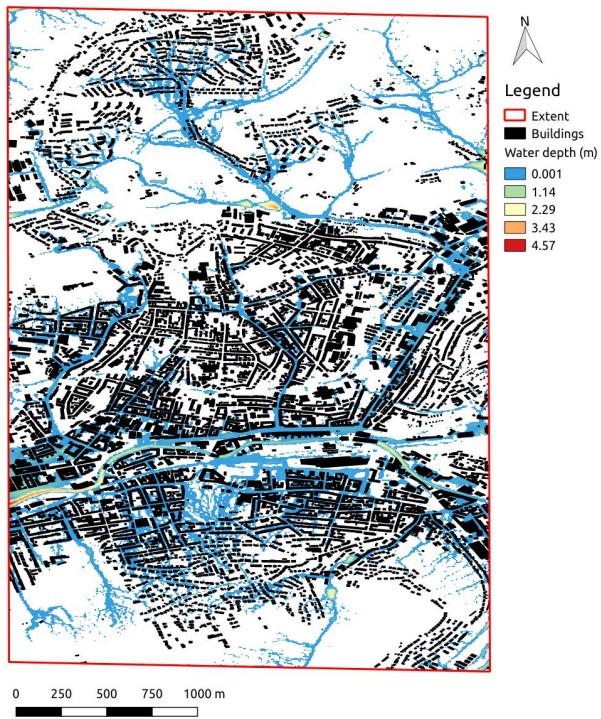

**Figure 6.** Water depth estimated by CAMC at the end of the simulation.

The comparison of ANUGA and CAMC considers the recommendations in [54,55]. The following model efficiency performance measures on all time series over the entire simulation domain were computed [54]: Nash–Sutcliffe Model Efficiency (NSE), Root Mean Square Error (RMSE) and Index of Agreement (IoA). Table 1 displays the formulation of the measures. Their spatial distribution is visualized in Figure 7 for water depth and in Figure 8 for velocity. The model efficiency performance measures consequently compare the agreement between the results of ANUGA and CAMC. Figures 7 and 8a show the NSE, which is a commonly used measure in hydrology. The RMSE in Figures 7 and 8b is in the same units as the data. The NSE and the RMSE use square root in their calculation leading to a potential bias when comparing models with offsets [55]. To compensate the potential shortcoming, the IoA in Figures 7 and 8c was included. Bias might be avoided by using absolute differences instead of square root. In Figures 7 and 8, green indicates very good or perfect agreement between ANUGA and CAMC. The evaluation is complemented by visually inspecting

the similarities and differences of the time series in form of residual plots, i.e., absolute error, at six gauge locations in Figures 9 and 10.

Regarding water depth in Figure 7, the reason that most of the area apparently shows very good agreement is that these cells were drained immediately after the cease of rainfall. Therefore, the time series produced by ANUGA and CAMC exhibit a high degree of resemblance. Considering only the cells that contained water at the end of the simulation, the mean NSE is nonetheless 0.61. The mean RMSE is 0.39 m, while the mean IoA is 0.65. This improves even when regarding the median. The model performance is summarized in Table 2. A general source of the disagreement is that ANUGA simulates the runoff flows on the basis of a triangular mesh. Hence, the results had to be transformed into raster data for comparison. The NSE is sensitive to outliers. It is visible in Figure 7a, where many cells are marked by a great difference. Regarding Figure 6, cells with lower water content tended to underperform. The reason might be that the dynamic of runoff flows in these cells were higher, and thus produced more mismatches. The situation is more nuanced in Figure 7b,c. The RMSE indicates underperformance where cells were less dynamic but contained more water. The IoA resembles more the NSE in that it suffered from more dynamic cells. On the other hand, the disagreement is less severe, which might be caused by the absence of a square root in its formulation.

In the case of velocity, the agreement between ANUGA and CAMC is considerably weaker. Considering the cells containing water at the end of the simulation (Figure 6), Table 2 shows that the mean for NSE is 0.34, for RMSE is 0.13 m/s and for IoA is 0.39. The median for the three measures is slightly better. All three performance measures in Figure 8 illustrate discrepancies between the models at the main pathways of the runoff flows where the hydrodynamics are very strong. This situation is exacerbated by the dense urban fabric, which cancel the flows. The extent of discrepancy in the model performance is caused by the continuous temporal variations. The fundamental reason is that in diffusive-like models, such as CA, inertia terms and momentum conservation are neglected. The reduction of model complexity through the omission of momentum as the driving force is in the nature of a simplified model like CA. It is a trade-off to achieve high computational efficiency to fulfill its objectives. Moreover, the possibility of discharging water in all neighboring cells requires an aggregation of the velocities in order to render them comparable to the unidirectional velocity vector that is produced by ANUGA. This implementation is another source of discrepancies. In the rare cases that other CA studies compare velocity, the results are similar [35]. The velocity estimation of CAMC can only serve as an indication.

**Table 2.** Summary of model performance for cells containing water at the end of the simulation (Figure 6).

|  |  | NSE | RMSE | IoA |
|---|---|---|---|---|
| Water depth | Mean | 0.61 | 0.39 m | 0.65 |
|  | Median | 0.67 | 0.25 m | 0.67 |
| Velocity | Mean | 0.34 | 0.13 ms$^{-1}$ | 0.39 |
|  | Median | 0.38 | 0.11 ms$^{-1}$ | 0.42 |

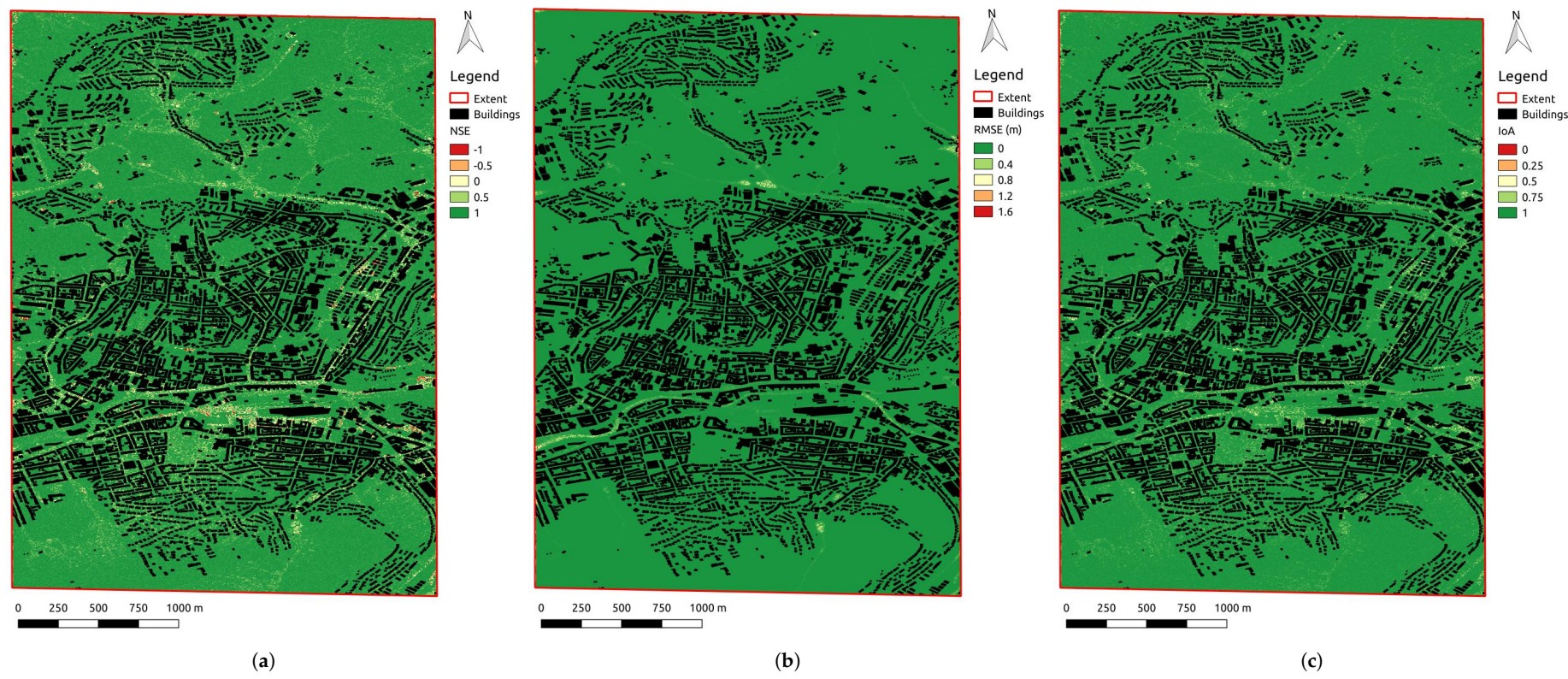

(**a**)　　　　　　　　　　　　　　　　　(**b**)　　　　　　　　　　　　　　　　　(**c**)

**Figure 7.** Spatial distribution of model performance for water depth. Green indicates very good or perfect agreement. (**a**) NSE; (**b**) RMSE; (**c**) IoA.

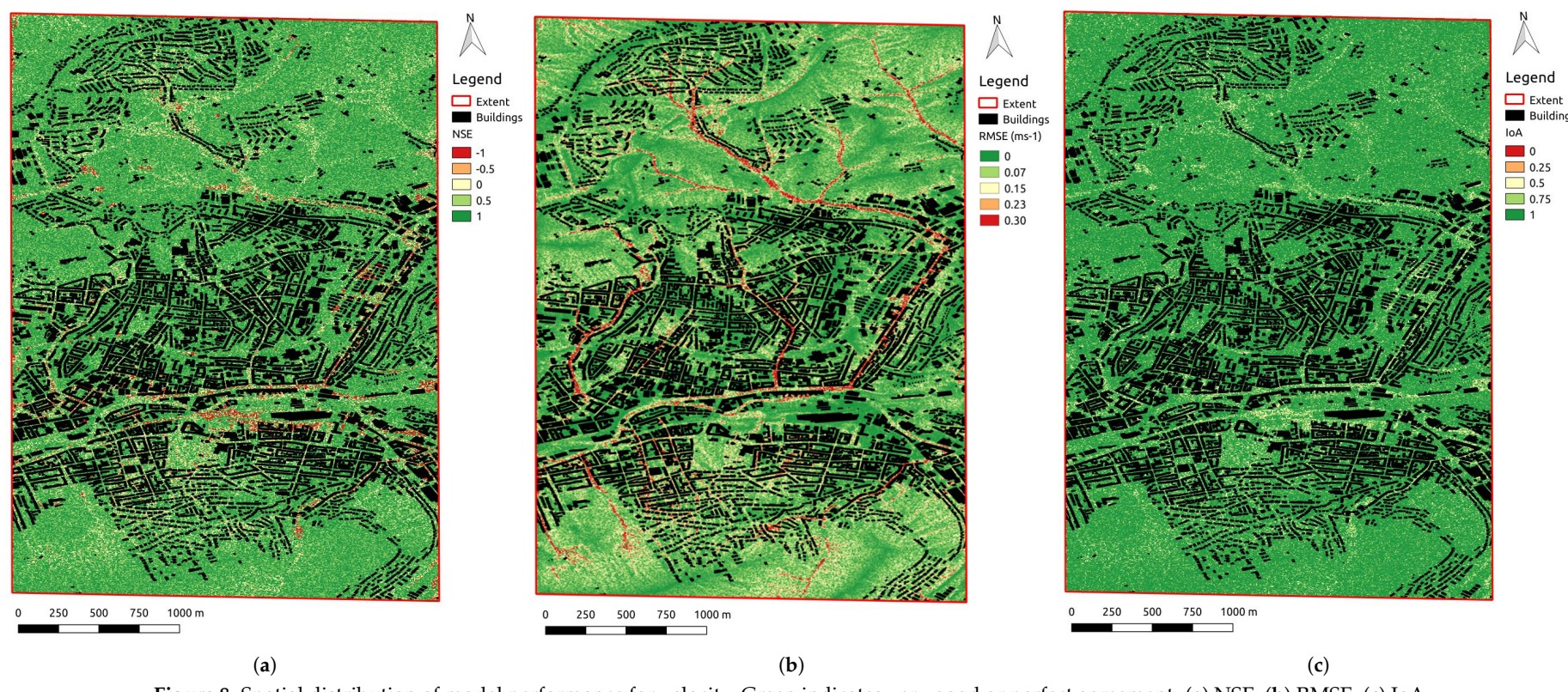

**Figure 8.** Spatial distribution of model performance for velocity. Green indicates very good or perfect agreement. (**a**) NSE; (**b**) RMSE; (**c**) IoA.

The residual plots in Figures 9 and 10 serve to illustrate the difference in the temporal development at the six gauge locations marked in Figure 4. The differences between the water depth estimations in all locations increased during rainfall and stayed mostly constant after its end. Gauge 6 even exhibits a relatively great reduction but on the lower level, while the highest difference is at gauge 2. It is not clear whether there is a tendency to over- or underestimate. Both behaviors are visible. Gauges 3 and 6 depict the smallest difference. Both locations are merely in the pathway of runoff flows. Gauges 1 and 2 are situated at the limit of the simulation domain and suffer from high differences between the water depth estimations. The cause might be the reflective boundary definition, which does not allow water flows to leave the simulation domain, leading to a general overestimation of water depth. Gauges 4 and 5 are ponding areas well within the simulation domain. They witness moderate differences but in the opposite directions (underestimation of CAMC at gauge 4, while overestimation at gauge 5). The differences in the estimation of velocity are stronger and more dynamic. As in the estimation of water depth, the differences raises with the start of rainfall. However, in contrast to water depth, the difference changes considerably at gauges 2, 3 and 5 over the whole time series. Such strong temporal variations contribute to the weak agreement between both models in the estimation of velocity. At gauge 3, the estimation even changed from the underestimation of CAMC to overestimation. The reason that the differences in the velocity estimation converge is related to the decreasing amount of momentum in the system at the end of the simulation. It allows us to suggest that CAMC performs better in the estimation of lower velocities, which is supported by the better agreement in Figure 8 of cells losing water. The introduction of a threshold for negligible water flows and a mechanism ensuring equal or higher water depth in central cells than neighboring cells seems to be successful in controlling oscillation, which commonly inflicts CA models [35].

At the resolution of 6 m, ANUGA required around 7.5 h to complete the simulation, whereas CAMC required 6 h. However, the simulation of ANUGA employed 4 CPU cores. On the same computer, CAMC run on one core.

A general limitation concerns the lack of real-world measured data. To the knowledge of the authors, no comprehensive data set on urban inundation exists. The only options are to use laboratory-measured or simulated data. The consequence is that other factors might influence the model performance. For instance, the velocity of discharge to all neighboring cells needs to be aggregated in order to be compared to the unidirectional velocity vector produced by ANUGA. Moreover, ANUGA uses triangular tessellation, while CAMC relies on square tessellation. That required to transform the results of ANUGA into a raster data set. A related problem that concerns CAMC is that buildings need to be approximated with square cells. In a dense urban area, that might have a non-negligible impact. Two solutions, either combined or alone, are conceivable. One is to diversify the shapes of cells (e.g., triangular and hexagonal) and neighborhoods (e.g., Moore neighborhood); another is to introduce a variable cell size. A variable cell size would enable refinement of the mesh at critical interfaces, such as river banks and buildings. CAMC also suffers from numerical instability causing oscillation in the water distribution. Although a threshold for negligible water flow and a mechanism countering higher water depth in neighboring cells than the central cell were introduced, the instability might still influence the model performance. The inclusion of an inertia factor might remedy this behavior. In this regard, CAMC suffers from the same weakness as diffusive wave models, which is particularly visible in the estimation of velocity. Furthermore, the extension of CAMC with a 1D model component would benefit its applicability in stormwater planning by enabling the simulation of river flows and sewer networks. Finally, although CAMC is fast, the capacity for parallel computation is presently a common feature and should be supported by CAMC.

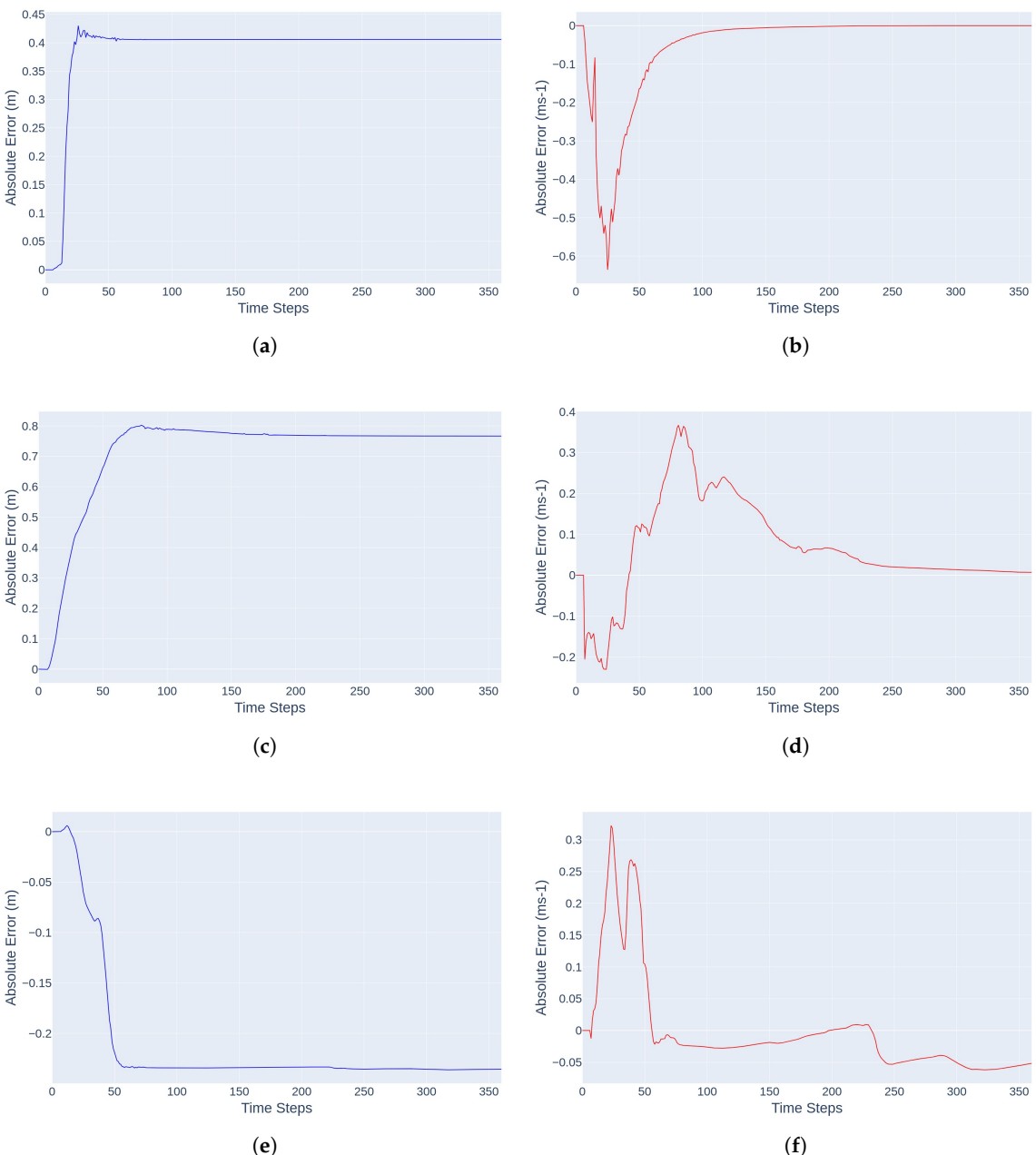

**Figure 9.** Residual plots (ANUGA-CAMC) for water depth (**left**) and velocity (**right**): (**a**,**c**,**e**) Water depth at gauges 1–3; (**b**,**d**,**f**) Velocity at gauges 1–3.

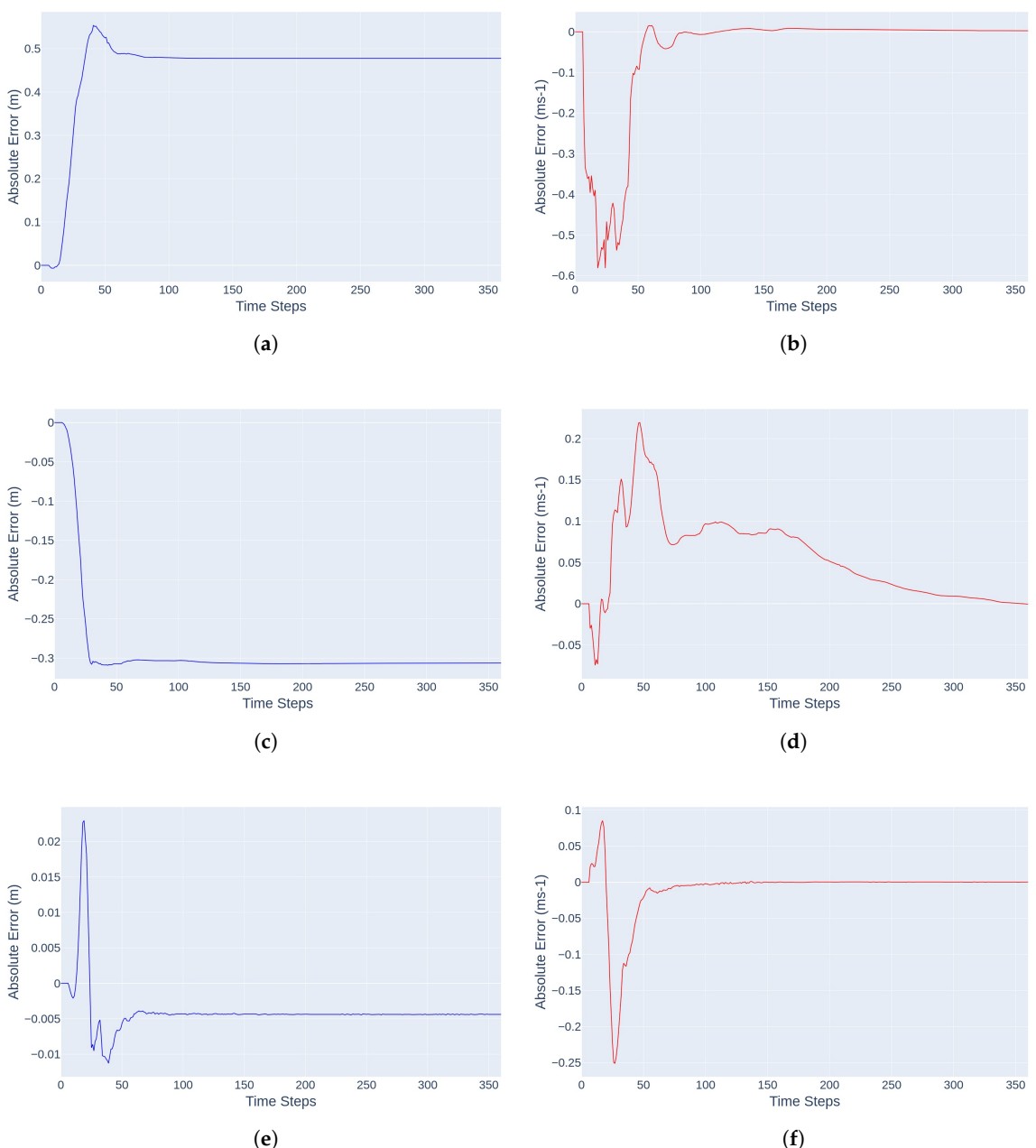

**Figure 10.** Residual plots (ANUGA-CAMC) for water depth (**left**) and velocity (**right**): (**a**,**c**,**e**) Water depth at gauges 4–6; (**b**,**d**,**f**) Velocity at gauges 4–6.

## 5. Conclusions

In conclusion, simple matrix and logic operations implemented in CAMC achieve acceptable performance in the estimation of water depth comparing to the computationally expansive SWE-model ANUGA, with an average NSE of 0.61, RMSE of 0.39 m and IoA of 0.65. The good agreement was achieved in a densely built-up area with about 12,000 buildings. Other CA models disregarded the effects which buildings could have on the flow characteristics, likely because of the involved effort for model set-up and excess runtime. In contrast, the estimation of velocity perform considerably less well, with an average NSE of 0.34, RMSE of 0.13 m/s and IoA of 0.39. However, because of the ignorance of momentum conservation in diffusive wave-like models like CA and the possible multi-directional discharge of water, CA models are inherently disadvantageous to estimate velocity

vectors. The consequence is that CA models are less applicable for the design of flood defense and structural damage assessment because flow velocities and incidence angles affect the structural integrity of the exposed structures [56]. Such limitations, however, is a trade-off of simplified models to realize their specific advantage. The high computational efficiency obtained by CAMC is its advantage. It only required 6 h on one CPU to simulate a densely built-up area at the spatial resolution of 6 m. The SWE model ANUGA in turn needed 7.5 h with four CPUs. This computational performance is useful in the estimation of inundation with high spatial and temporal resolution or short runtimes. In this study, the short runtimes and its flexible implementation in Python render it possible that CAMC is embedded into a "live" urban flood warning system [40] with web visualization of the output at every time step. Subsequent research will focus on realizing the potential of Motion Cost fields by using infiltration and evapotranspiration in the planning and optimizing of stormwater runoff source control. Its extension and coupling with other models, such as drainage networks and rainwater harvesting, are conceivable.

**Author Contributions:** Conceptualization, M.I.; methodology, M.I.; software, M.I.; validation, M.I.; formal analysis, M.I.; investigation, M.I.; resources, M.I.; data curation, M.I.; writing—original draft preparation, M.I.; writing—review and editing, M.I.; visualization, M.I.; supervision, F.-J.C., H.J.; project administration, F.-J.C., H.J.; funding acquisition, F.-J.C., H.J. All authors have read and agreed to the published version of the manuscript.

**Funding:** This research and the APC was funded by the Ministry of Science and Technology, Taiwan (Grant number: 107-2621-M-002-004-MY3).

**Acknowledgments:** We are grateful to the Geobasis NRW for providing data. Map data copyrighted OpenStreetMap contributors and available from https://www.openstreetmap.org. The authors would like to acknowledge the funding provided by the Tsinghua University-Veolia Environment Joint Research Center for Advanced Environmental Technology. The authors thank the Editors and anonymous Reviewers for their constructive comments that are greatly contributive to the revision of the manuscript.

**Conflicts of Interest:** The authors declare no conflict of interest. The funders had no role in the design of the study; in the collection, analyses, or interpretation of data; in the writing of the manuscript, or in the decision to publish the results.

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
