# Peer review of "Efficient Urban Inundation Model for Live Flood Forecasting with Cellular Automata and Motion Cost Fields"

_water, doi:10.3390/w12071997_

Round 1

Reviewer 1 Report

(1) The abstract may be improved by adding some core finding and the real significance 

(2) Evaluation criteria for the model performance should be detailed in the Methods section 

(3) The results are not adequately discussed to highlight the novelty and significance of the manuscript. Further improvement to the discussion is recommended to aid readers

(4) The conclusions should be revised to focus on the core findings (quantitative) and its real significance 

Reviewer 2 Report

Manuscript of the title "Efficient Urban Inundation Model for Live Forecasting with Cellular Automata and Motion Cost Fields" by M. Issermann, F.J. Chang and H. Jia.
The paper deals with an urban inundation model based on cellular automata (CA) in combination with Motion Cost fields (MC).
The model is applied to an urban area in the German city of Wuppertal and results are validated by simulations obtained by the hydrodynamic modelling tool ANUGA.
The simplified mathematical model is faster than ANUGA and it seems to be adequately accurate, although a thorough analysis of the comparison between the results of the two models is required.

General comments:

-In the Introduction differences between CA e CAMC should be stressed.
At page 2 lines 66-68 the authors should state clearly that CAMC approach may consider infiltration and evapotranspiration, otherwise these sentences are unclear.
- The authors should explain the physical meaning of equation 2. For instance, the depth of unsaturated soil is assumed to be constant in time. I think that this assumption needs to be justified.
- The description of the mechanism by which water flows into other cells according to MC is too concise (lines 122-123).
The authors should explain in detail the procedure of attributing the discharge of the central cell , maybe with an example.
- The presence of buildings in an urban flood concerns not only the exchange of inflows, but also the hydrodynamics induced around the impacted building.
The hydrodinamics is affected by velocities and incidence angles of the flow hiting the building.
Moreover, the existence of neighborhoods and close buildings lead to a larger super-elevation with respect to isolated structures (e.g. Postacchini et al., Flood impact on masonry buildings: The effect of flow characteristics and incidence angle, Journal of Fluids and Structures, 88, 48-70, 2019).
These aspects cannot be considered in simplified simulations but they can be significant in the assessment of flood risk for people and structures.
- The comparison between the results obtained with the two models could be deepen and a comparison between the flow velocities could be useful. Beside the water level also the flow velocity is a fundamental aspect in the urban area flooding.
- Do the results of Figure 7 (where units of measure are missing) refer to the absolute or relative error between the two solutions?
It is important for understanding the oder of magnitude of the error in the evaluation of the water level.
Differences in water level evaluation have significant impacts on the risk assessment of pedestrians flood-induced evacuation (e.g. Bernardini et al., Flooding Pedestrians’ Evacuation in Historical Urban Scenario: A Tool for Risk Assessment Including Human Behaviors, Springer, 2019).

Some specific comments:
page 1 line 9 Replace "both" with "the two"
page 1 line 18 Replace "diving" with "driving"
page 1 line 26 Replace "shedding" with "smoothing"
page 2 lines 40-41 It should be "Their low computational efficiency restricts their usage..."
Figure 3. The gauge numbers are too small.
page 10 line 227 Replace "an relatively" with "a relatively"
Figure 7. Please, add the unit of measure
pag 11 line 252 Please, replace "render" with "renders"
pag 11 line 254 Please, add "that" between "the inlfuence" and "buildings"
pag 11 line 261 replace "author" with "authors"
pag 12 lines 309-310 Please, add details about this reference

Reviewer 3 Report

Referee report on manuscript «Forecasting with Cellular Automata and Motion Cost Fields» by Issermann et al.

The manuscript presents a cellular automata model for simulating surface water floods in an urban environment. The main aim is to develop a computationally fast model to enable live simulation. The model has been applied to a case study and compared with a hydrodynamic model. The main interesting feature is the approach of introducing motion cost fields in the CA model.

The manuscript is well written and provides interesting insights in a modelling technique. I have a few remarks and suggestions for improving the manuscript before publication.

-It is not described how the soil properties are taken into account. Please add the units of roughness etc.

-Please comment on the restriction of the boundary conditions. As water is not allowed to leave the simulation domain, the cells near the borders must have overestimated water levels.

-The figures 3, 5, 6 are not orthogonal. Please rectify the maps. Moreover, it is not clear if these maps are a 2D or a 3D visualization. Please add a north arrow or a coordinate system.

-It is not clear how the difference (figure 7) is calculated (CAMC – ANUGA or ANUGA – CAMC). In an case, a difference of 0.4 m (I suppose these are m of flood depths at gauge location – it is not clear from figure 7) is a huge difference. This huge error has to be discussed in detail.

- IN the conclusions, the author write that MC provide an interesting alternative as transition rule. However, the authors did not compare MC with another approach. Thus, this conclusion cannot be drawn from the study setup.

Author Response

lease see the attachment.

Round 2

Reviewer 1 Report

The revised manuscript sufficiently addresses the review comments 

Author Response

Thank you.

Reviewer 2 Report

I carefully read the revised version of the manuscript, the authors answered all the questions and they addressed my concerns. I have only one minor comment:

Abstract at line 13: the sentence "but differ in the area of disagreement" is too concise and not clear. The authors should better explain this matter.

Author Response

We want to express our gratitude to the Editors and Reviewers in their continues effort. The issue was addressed and we hope that it meets the expectations. The revised manuscript with highlighted changes is attached. Thank you very much for your support.

I carefully read the revised version of the manuscript, the authors answered all the questions and they addressed my concerns. I have only one minor comment:

Abstract at line 13: the sentence "but differ in the area of disagreement" is too concise and not clear. The authors should better explain this matter.

Reponse: Thank you for this remark. The concerning phrase refers to the issue that the model performance measures (Nash-Sutcliffe Model Efficiency, Root Mean Square Error, Index of Agreement) indicate different areas in the simulation domain where the estimations produced by CAMC and ANUGA do not correspond well. We mentioned the issue in the result section [262-268]. The abstract was also modified within its limits to highlight this issue.
